# Predicting Potential Distribution of *Teinopalpus aureus* Integrated Multiple Factors and Its Threatened Status Assessment

**DOI:** 10.3390/insects15110879

**Published:** 2024-11-09

**Authors:** Congcong Du, Xueyu Feng, Zhilin Chen, Gexia Qiao

**Affiliations:** 1Key Laboratory of Zoological Systematics and Evolution, Institute of Zoology, Chinese Academy of Sciences, Beijing 100101, China; cleverduwang@gmail.com; 2Key Laboratory of Ecology of Rare & Endangered Species & Environmental Protection, Guangxi Normal University, Ministry of Education, Guilin 541006, China; fengxueyu66@163.com (X.F.); chenzhilin35@163.com (Z.C.); 3Guangxi Key Laboratory of Rare & Endangered Animal Ecology, Guangxi Normal University, Guilin 541006, China; 4College of Life Science, Guangxi Normal University, Guilin 541006, China

**Keywords:** biotic interaction, climate change, expert map, host specificity, SDMs

## Abstract

Based on the “Hutchinson niche hypothesis” and the “Eltonian Noise Hypothesis” about ecological niches, as well as the “resource dependent hypothesis” about biotic interactions, this study integrated climate data, host plants, and empirical expert maps to predict potential distributions based on the Maxent and random forest (RF) models. Utilizing the species richness of host plants as a surrogate for biotic interactions was a simple and effective way to significantly improve the predictive performance of Species Distribution Models (SDMs). Given climate change, their distribution is significantly shrinking, increasing their threatened level in the future. Abiotic factors could not only directly affect the distribution of *T. aureus* but also indirectly impact it through host plants. This was evident in the delayed response of *T. aureus* to climate change compared to its host plants, which is called the “hysteresis effect” caused by biotic interactions. Considering its narrow altitude range, vulnerability to climate change, host plant specialization, and the survival status of its highly dependent host plants, we recommended designating *T. aureus* as currently vulnerable.

## 1. Introduction

Predicting the niche and potential distribution of a species is essential for biodiversity research [1]. Species distribution models (SDMs) can associate the known distribution occurrences of organisms with environmental factors to foresee their potential geographical distribution in a specific time and space [2], and it is widely used for discussing the response of key species in the ecosystem under the background of global climate change [3,4], the potential distribution of invasive species [5], the protection of endangered and rare species, and the planning of protected areas [6,7].

The distribution of a species is a complex expression of their ecological and evolutionary history, which is controlled by numerous factors and different intensities at different time and space scales [2]. On a shorter time scale, environmental factors (climate, landform, soil, physiological environment, etc.), biological factors (biotic interaction), and the dispersal ability of the species are three crucial factors affecting the species distribution [8,9]. The “Hutchinson niche hypothesis” states that the abiotic conditions are the ones limiting species distribution at large spatial resolutions and extents [9]. Therefore, current SDM researchers mostly analyzed data based on abiotic environmental factors [3,10]. However, despite the “Eltonian Noise Hypothesis,” which suggests that biotic variables are less relevant to species distribution at large geographic scales [9], biotic interactions have increasingly been acknowledged to also play a pivotal role, even at a regional scale [11,12,13,14,15]. Therefore, considering the biotic interactions in the SDMs can enhance prediction accuracy [16,17,18,19,20], which is of great significance for protected species with niche specialization [15].

*Teinopalpus aureus* Mell, 1923 is a swallowtail butterfly species (Papilionidae: Lepidoptera) endemic to the high elevation of mountains in South China, Vietnam, and Laos [21,22,23]. Although rarely encountered in nature, it receives close attention from entomologists, environmentalists, and the public because of its striking appearance, and it is known as a flagship species [24]. In China, it is proposed as the national butterfly and under the first-class state protection [25]. As a flagship species, the conservation of *T. aureus* also supports the protection of other ecological communities in the same area. However, its conservation status is data deficient (DD) on the IUCN Red List of Threatened Species due to a lack of population and distribution information [26,27]. Recently, it has been reported to be found in many new places [28,29,30], but its potential whole distribution is still unknown, which was a great obstacle to its effective scientific protection and management. Xing et al. predicted its suitable habitats, taking both climate suitability and habitat type into account by refining its distribution map to areas with forest cover information [31]. Despite its current occurrence seeming to depend on evergreen broad-leaved forests, it showed a high host specialization, mainly with hosts on some species of Magnoliaceae, especially during the egg and larval stages [24]. The “resource dependent hypothesis” states that the intricate relationship with host plants for resources might constrain their species diversity patterns from being positively correlated to specialist herbivorous insects [32]. Wang et al. found that this butterfly exclusively occurred in its preferred primeval broadleaf forests, mainly driven by the specific larval host plants (i.e., three Magnoliaceae species) [24]. Many scholars have already reviewed, advocated for, and attempt to incorporate biotic interactions into SDMs [15,19,33]. Thus, it can be speculated that integrating real host plant information into SDMs could enhance their prediction performance, and we could accurately predict the distribution of *T. aureus*.

It is worth noting that occurrence records typically provide fine-scale occurrence information that could be characterized by its environmental association, but they usually suffer from observer biases and do not representatively or fully address the geographical or environmental range occupied by a species [34]. In contrast, expert maps, which outline the geographical region where a species is believed to occur [34,35], appear particularly good at delineating range edges beyond which a species is unlikely to occur because of its emphasis on avoiding omission errors (false negatives) [36,37]. At coarse resolutions, the boundaries of the expert maps were reasonably accurate [34,35,38]. Therefore, combining the complementary informative attributes of both data types can improve fine-scale, large-extent predictions [34]. Mainali et al. also pointed out that expert maps can help to reduce SDM omission errors from the outset by limiting the boundaries of potential space [39].

Borgelt et al. found that 85% of DD amphibians are likely under threat of extinction, and the proportion also exceeds half in mammals and reptiles [40]. In this way, against the backdrop of a sharp decline in global insect diversity and abundance [41], their threatened status may become even more critical. Without a doubt, taking prompt protective measures is key to maintaining insect species and populations [42]. *T. aureus* is sensitive to environmental temperature, and habitat loss and fragmentation could be major threats to the species [43,44]. Potentially facing multiple threats, the extinction risk of *T. aureus* has not been assessed yet. Precise geographical range information is one of the important bases for assessing the endangered status of a species [45]. Therefore, in this study, by integrating occurrence data and expert maps, along with abiotic and biotic interaction factors, the current and future potential distribution areas of *T. aureus* are predicted to explore the following questions: (1) Can biotic interactions and expert maps enhance the predictive performance of the SDMs? (2) How will the suitable habitats of *T. aureus* respond to climate change? (3) What is the threatened status of *T. aureus*? The answers will help to reveal the survival status of *T. aureus*, providing a basis for its rational protection and scientific management.

## 2. Materials and Methods

### 2.1. Occurrence Data

Following the method of Xing et al. [31], we re-proofread, collected, and updated the occurrence data of *T. aureus* based on the Global Biodiversity Information Facility (https://www.gbif.org, accessed on 15 March 2023), scientific publications, museum specimen records, and field observations. In total, 57 occurrence points were collected from China, Vietnam, and Laos (Figure 1a; Appendix A). The spatial distance between each site was large enough to somewhat reduce the potential for spatial autocorrelation effects.

Based on current research [24,43,46,47] and our own fieldwork, 6 host plants of *T. aureus* (*Parakmeria nitida* (W. W. Sm.) Y. W. Law, *Michelia foveolata* Merr. ex Dandy, *Michelia chapensis* Dandy, *Parakmeria lotungensis* (Chun & C. H. Tsoong) Y. W. Law, *Manglietia fordiana* Oliv., and *Michelia maudiae* Dunn) were accurately recorded in the egg and larval stages. Therefore, the occurrence points data of these 6 host plants was collected from the GBIF. To avoid potential spatial autocorrelation to some extent, only one record within the same grid (2.5 × 2.5 arc min, 4.6 km resolution at the equator) was retained. Ultimately, 1008 occurrence points (ranging from 18 to 400) were utilized in the subsequent SDMs (Figure 1a).

### 2.2. Expert Map

The recent expert map of *T. aureus* was mapped by Igarashi [23]. Considering recent discoveries of numerous new distribution records of *T. aureus*, especially in China, we updated the expert map (Appendix A) based on the latest collected occurrence records and the old expert map by Igarashi [23]. Of course, the creation of accurate expert maps relies on various evidence and empirical knowledge [39]. However, despite the disputed accuracy of our updated expert map, it still holds significant value at larger spatial scales and coarser resolutions, which constrain predictions of a species distribution model parameterized with incidental point occurrence records [34].

### 2.3. Abiotic Variables

The contemporary and future 19 environmental climatic variables representing temperature and precipitation were downloaded at a spatial resolution of 2.5 arc min from the WorldClim dataset (Global Climate Data, http://www.worldclim.org/, accessed on 15 March 2023) [48]. The current bioclimatic variables represent averages of a 50 yr period from 1950 to 2000. Four future climate scenarios (SSP126, SSP245, SSP370, and SSP585) were selected, which represented the distinct probable future greenhouse gas concentrations and were generated based on downscaled global climate model (GCM) data from CMIP6 (IPPC Sixth Assessment). Every future climate scenario covered four different periods averaged over 20 yr periods: 2030 (average for the years 2021–2040), 2050 (average for the years 2041–2060), 2070 (average for the years 2061–2180), and 2090 (average for the years 2081–2100).

In addition, considering that *T. aureus* inhabited high-altitude mountainous regions, global elevation data representing topographical heterogeneity (TOPO) were also selected to represent abiotic variables, and they were also downloaded at a spatial resolution of 2.5 arc minutes from the Global Map data archives (https://globalmaps.github.io/, accessed on 15 March 2023).

### 2.4. Biotic Variables

How to appropriately incorporate biotic interactions into SDMs in a quantitative way is a critically important question. If occurrence or abundance patterns of partners are symmetrical, joint SDMs (jSDMs) can account for multiple species (assumed biotic) interactions, environmental covariates, species traits, and phylogenetic relationships through a hierarchical Bayesian modelling approach [49,50]. However, this assumption may not be adequate for all species’ relationships, particularly for herbivore interactions. In such cases, according to the “resource specialization hypothesis” [32], areas with high abundance or species diversity of host plants tend to have a higher probability of distribution or species diversity of specialized herbivorous insects. Therefore, for specialized herbivorous insects, incorporating the species diversity pattern of hosts as a biotic variable to be a predictor in SDMs might be a simple yet effective method [15,18,20].

As the occurrence data were available in the form of presence-only records, we employed the maximum entropy method (hereafter Maxent) [51] for conducting SDM analyses. For each host plant of *T. aureus*, we conducted a pre-Maxent analysis and combined its occurrence points with all 19 climatic variables and elevation data. The training data were 75% of the sample distribution points chosen randomly, and the remaining 25% were designated as testing data. Based on the jackknife method, the contribution value of each climate variable was evaluated. There often exists a strong correlation between climate variables, which may lead to overfitting of the model [52,53]. Therefore, after conducting a correlation analysis of 19 climate variables, for the highly correlated variables (Spearman correlation coefficient > 0.8), only the ones with higher contribution values in the pre-Maxent analysis were retained for SDM analyses again. To ensure that the possibility of these species’ distribution appeared close to normal, the package “ENMeval” was used in R to optimize the Maxent model for further reducing the overfitting [54]. The Maxent model was executed with a maximum number of iterations of 10,000, the 10–5 default convergence threshold, and 100 replicates under bootstrap run type to guarantee the model’s accuracy [55]. Based on the IPCC’s classification criteria for likelihoods, the output results were resampled by ArcGIS 10.2 (ESRI, Inc., Redlands, CA, USA). Using 0.66 (likely) as the threshold, the distribution probability was converted into presence or absence raster; thus, the distribution map for each host plant was obtained. Then, the distribution maps of six total host plants were overlaid to obtain their distribution patterns of species richness, which are regarded as predictive factors of *T. aureus* representing biotic variables.

### 2.5. Test of Climatic Niche

Firstly, the climatic spaces of *T. aureus* and its host plants were created following the method of Ponti et al. [56]. Then, species data were projected onto the first two axes of a principal components analysis (PCA) in the climatic space, and the species occurrence density was calculated using a kernel smoother in the gridded PCA environmental space at a resolution of 1000 × 1000 cells, referred to as PCA-env in Broennimann et al. [57]. Next, the statistical tests on similarity of climatic niches between *T. aureus* and its host plants were conducted using the random test method, which could quantify the dynamics of climate niche into three parts: stability, expansion, and unfilling [57,58]. Stability was defined as the proportion of niche space occupied by both partners, while expansion was defined as the proportion of niche space occupied only by host plants and unfilling was defined as the proportion of niche space occupied only by *T. aureus*. If there was a high similarity and strong stability in the climatic niches of both partners, it further indicated their close relationship, suggesting that the host plants could serve as a biotic variable for predicting the habitat suitability of *T. aureus*. All niche dynamics analyses were performed in the “ecospat” R package [59].

### 2.6. Species Distributions Models

To explore the influence of expert maps and host plants, four different combinations of predictive variables were analyzed: (1) climatic and elevation data, (2) climatic, elevation, and expert map data, (3) climatic, elevation, and species richness of host plants data, (4) climatic, elevation, expert map, and species richness of host plants data. All SDM analyses were conducted using the maximum entropy method [51], because it is constructed using only presence data and has good performance even with small sample sizes [60,61]. Nevertheless, in order to avoid the unreliability of a single model, a comparative analysis was also conducted using the random forest (RF) model. To integrate expert maps into our Maxent models, the best offset layer was selected as a bias file in Maxent. Following the method of Merow et al. [34], various parameters (prob = 0.5, 0.7, 0.9; rate = 0, 0.01, 0.1, 10; skew = 0.5) were used to describe different decay curves across the transect to produce a total of 12 offsets, which were calculated in R package “bossMAPs”. Then, the best offset layer (prob = 0.9, rate = 0.1, and skew = 0.5) with lowest Akaike information criterion (AIC = 2319.2) value was selected to constrain predictions of SDMs. The selection of climate variables and the execution of the Maxent model followed the above prediction of potential distribution zones of host plants. Similarly, based on the IPCC’s classification criteria for likelihoods, the output result was resampled by ArcGIS 10.2 (ESRI, Inc.) into four categories: unsuitable area (probability: 0–0.33), low suitable area (probability: 0.33–0.66), medium suitable area (probability: 0.66–0.90) and high suitable area (probability: 0.90–1.00).

Then, following the method of Han et al. [62], the random cross validation approach was used to evaluate the predictive performance of the model. We estimated model performance using the area under the receiver operator characteristic curve (AUC, ranging from 0 to 1), true skill statistic (TSS, ranging from −1 to +1), Boyce index (ranging from −1 to +1), odds ratio skill score (ORSS, ranging from −1 to +1), and symmetric extremal dependence index (SEDI, ranging from −1 to +1) [63,64,65,66,67,68]. We compared the predictive performance of the SDMs (Maxent and RF models) with the above four combinations of predictive variables using the two-tailed Wilcoxon signed-rank test.

The predictions in four future periods (2030, 2050, 2070, and 2090) under different greenhouse gas concentration scenarios (SSP126, SSP245, SSP370, and SSP585) were also conducted. Finally, we applied a jackknife procedure to evaluate the relative importance of each of the predictor variables to correctly predict new ranges in current and future models [69]. Given that the predictive variables might also be correlated, structural equation models (SEMs) were further used to test for direct and indirect effects of abiotic variables on the distribution probability of *T. aureus* using the R package “piecewiseSEM” [70].

### 2.7. Threatened Status Assessment

The contemporary threatened status of *T. aureus* was assessed based on the occurrence records and predicted presence/absence raster data, separately. The value of extent of occurrence (EOO) and the area of occupancy (AOO) were measured and calculated according to the IUCN Red List Criteria in accordance with the categories and criteria established by the IUCN Red List of Threatened Species. Furthermore, based on the predicted future presence/absence raster data, the value of future extent of occurrence (EOO) and the area of occupancy (AOO) were also measured and calculated in four future periods under different greenhouse gas concentration scenarios. The corresponding analysis was conducted using package “red” in R.

## 3. Results

### 3.1. Comparison of Climatic Niche Between T. aureus and Host Plants

The current niche overlap and similarity between *T. aureus* and its host plants were both very high (Schoener’s *D* = 0.63, *p* = 0.19; Hellinger distance *I* = 0.85, *p* = 0.18). Niche dynamics analyses suggested that their stable niche was also very high (91.61%) (Figure 1b).

### 3.2. Influence of Expert Map and Host Plants on SDMs

The current potential distribution of *T. aureus* based on four different combinations of predictive variables varied noticeably under both the Maxent and RF models (Appendix A). Compared to predictive models that only include climate and altitude, the addition of the expert map did not result in significant changes in the Maxent model, while it showed a significant improvement in the RF model (TSS and SEDI: *p* < 0.001; ORSS: *p* < 0.05, two-tailed Wilcoxon signed-rank test) (Figure 2). In comparison to predictive models that included climate, elevation, and host plants, the predictive power of the SDMs was also significantly improved after incorporating the expert map (Maxent model: Boyce, *p* < 0.001; RF model: TSS and SEDI: *p* < 0.001, two-tailed Wilcoxon signed-rank test) (Figure 2). Meanwhile, the inclusion of the host plants also led to a considerable improvement; particularly notable was the significant enhancement observed in some assessment metrics based on the Maxent (AUC and Boyce: *p* < 0.001, two-tailed Wilcoxon signed-rank test) and RF (TSS and SEDI: *p* < 0.05, two-tailed Wilcoxon signed-rank test) models (Figure 2). Therefore, after adding the host plants and expert map, the predictive performance of the SDMs improved to some extent, although there was no consistency among the different evaluation parameters. Overall, the models incorporating all the predictive variables showed the highest value of different assessment metrics (especially AUC, Boyce, ORSS, and SEDI in the Maxent model; TSS, SEDI, and ORSS in the RF model) and demonstrated the best performance (Figure 2), and they were used in the following analysis.

Based on the jackknife analysis in the Maxent model, when the host plant variables were added to the predictive model, their contribution to the current potential suitable areas of *T. aureus* consistently ranked the highest and almost approached 50%. Additionally, in the predictive model incorporating the host plant variables, the temperature (bio4: Temperature Seasonality, bio9: Mean Temperature of Driest Quarter, bio2: Mean Diurnal Range) and elevation variables also demonstrated a significant contribution value (Appendix A). Similarly, the SEM analysis also indicated that the altitude, temperature, and precipitation significantly influenced the distribution probability of *T. aureus* and the species richness of its host plants. The species richness of the host plants not only significantly impacted the distribution probability of *T. aureus* but also emerged as the most influential variable (Figure 3).

### 3.3. Current and Future Distribution of T. aureus

Based on the predictive model with combined climatic, elevation, expert map, and species richness of host plants data, the current potential distribution zones of *T. aureus* exhibited a fragmented distribution pattern (Figure 4), especially under the Maxent model (Figure 4a). The high (probability > 0.90) suitability areas for *T. aureus* in China revealed by the Maxent and RF models were very similar, both mainly distributed in the Nanling Mountains located at the border of the Guangxi, Guangdong, Hunan, and Jiangxi provinces, and in the mountainous areas at the border of the Fujian, Zhejiang, and Jiangxi provinces, such as the Yandang Mountains, Jiufeng Mountains, and Wuyi Mountains, as well as in the Central Mountains of Taiwan Island (Figure 4). However, compared to the Maxent model, the RF model revealed a significant expansion in the area of medium and low suitability zones (Figure 4).

With climate change, the potential suitable habitat range of *T. aureus* under the Maxent model also gradually decreased under different greenhouse gas concentration scenarios (SSP126, SSP245, SSP370, and SSP585) (Figure 5; Appendix A), manifested by a decrease in remaining areas and an increase in shrinking regions both in and out of the current protected areas (Appendix A). The decreased areas were widely distributed within its potential habitat range and were particularly prominent on both the north and south sides of Vietnam as well as in the Hainan, Zhejiang, and Fujian provinces of China (Appendix A). Furthermore, there was no significant change in the future altitude niche of *T. aureus*, which remained consistent with the current situation, albeit with a markedly reduced probability (Appendix A).

### 3.4. Threatened Status

Based on the current occurrence records and predicted presence/absence raster data under the Maxent model, the AOO and EOO values of *T. aureus* were both very high (AOO > 2000 km^2^; EOO > 20,000 km^2^), indicating that it was in a state of Least Concern (LC). However, as the threshold for defining the presence of a species increased, the threatened extent of *T. aureus* intensified based on the AOO value. *T. aureus* was classified as in an endangered state (EN) only when the threshold was set at 0.90 (very likely) (Appendix A). Additionally, it is worth noting that the vast majority of the potential suitable range of *T. aureus* lay outside the protected areas (PAs) in China, with only a very small portion distributed within the PAs (Figure 4a; Appendix A). With climate change, the future threatened level of *T. aureus* increased. Particularly in the SSP370 scenario, its threatened level could even escalate to Critically Endangered (CR) status (Figure 6).

## 4. Discussion

### 4.1. The Power of an Expert Map and Biotic Variables for SDMs

Every species exists within a network of biotic interactions, spanning from specialization to generalization [71,72]. Generalist species access a wide range of biological resources, whereas specialized species can only utilize a limited portion of them [73]. Biotic interactions play crucial roles in species coexistence, formation, and extinction processes [72]. For herbivorous insects, host specificity (i.e., the degree of host dependence) may affect their distribution, with highly specialized species exhibiting distributional ranges that are enclosed within the range of their host [32,74,75]. In these cases, the climatic niche of the herbivorous species is expected to be constrained by that of the tree hosts [76]. The higher the host specificity, the more similar and the higher the overlap the climatic niche. Conversely, if the similarity and overlap of the climatic niches of the herbivorous insects and their host plants are higher, their interaction specificity is stronger [76], leading to a closer spatial distribution. In this study, the similarity and overlap of the current climatic niches of *T. aureus* and its host plants were very high (*D* = 0.63, *I* = 0.85, *p* > 0.005) and stable (91.61%) (Figure 1b), which indicated their close relationship and suggested the importance of host plants for *T. aureus* distribution.

Compared to predictive models that exclude host plants (climatic and elevation data, or climatic, elevation, and expert map data), the addition of host plants could improve all the evaluation metrics to some extent, being significant in the AUC, Boyce, ORSS, and SEDI in the Maxent model (Figure 2a) and in the TSS, SEDI, and ORSS in the RF model (Figure 2b). Therefore, these results clearly indicated that the host plants did enhance the predictive performance of the SDMs. Furthermore, incorporating the species diversity pattern of the hosts as a biotic variable to be a predictor in the SDMs was a simple yet effective method. Based on the resource dependence hypothesis [32], the distribution of *T. aureus* should be positively related to the species diversity pattern of its host plants. When host plant variables were added to the predictive model, their contribution to the current potential suitable areas of *T. aureus* consistently ranked the highest and almost approached 50% (Appendix A), which further confirms the importance of the host plants for *T. aureus*.

However, when considering the expert map, its performance in the RF model was better than that of the Maxent model. Compared to predictive models that only include the climate and altitude, the addition of the expert map did not result in significant changes in the Maxent model, while it showed a significant improvement in the RF model (TSS and SEDI: *p* < 0.001; ORSS: *p* < 0.05, two-tailed Wilcoxon signed-rank test) (Figure 2). The results indicated that the improvement in the predictive performance of the SDMs due to expert maps appeared to be model-dependent. Therefore, when and how expert maps should be used to stably and reliably enhance the predictive power of the SDMs is a question worth considering in the future.

### 4.2. Key Factors Determined the Distribution of T. aureus

The species richness of the host plants contributed the most to the current potential suitable areas of *T. aureus*, almost approaching 50%, implying its importance for the ecological survival of *T. aureus* (Appendix A). As an herbivorous insect, throughout the life cycle of *T. aureus*, from egg to larva and pupa stages, the host plants serve as its essential feeding and dependency resources [24,77]. The host plants identified for the eggs and larvae of *T. aureus* so far mainly belong to the family Magnoliaceae, exhibiting a high degree of host specialization. These host plants not only provide food for *T. aureus* but also serve as shelters for its survival, development, and reproduction, constituting vital components of its habitat. It is worth noting that these host plants all are endemic species with narrow distributions in southern East Asia, and they exhibit a fragmented distribution pattern (Appendix A). Among them, *Parakmeria nitida* and *Parakmeria lotungensis* are vulnerable (VU) species, while *Michelia chapensis* is a near threatened (NT) species, and all show decreasing distribution zones with climate change in the future (Appendix A). Although *Manglietia fordiana* is not listed in the IUCN conservation directory, the gradual reduction in its distribution with climate change indicates an unfavorable survival status, warranting attention from botanists. This indicates that the strong preference for host plants, as well as the rarity of the host plant species, the narrowness of their distribution, and their vulnerability to climate change, all could affect the survival status of *T. aureus*, making its outlook less optimistic.

Climate is the most fundamental factor influencing species distribution [9]. In addition to the host plants, the factors contributing most significantly to predicting the current potential suitable distribution of *T. aureus* mainly involve temperature-related factors, such as temperature seasonality (bio4) and the mean temperature of the wettest (bio8) and driest (bio9) quarters (Appendix A). For most animals, temperature often serves as a prerequisite factor influencing species distribution [78,79]. Every species possesses its unique phenology [80,81]. *T. aureus* reproduces twice a year, with the first generation occurring from early April to mid-May and the second generation occurring from late August to mid-September [77]. This life cycle is often dependent on annual climate fluctuations. Therefore, seasonal temperature changes have a significant impact on *T. aureus*. In southern China from April to September, it is typically the wet season [82], during which the temperature might determine the activity rhythm of *T. aureus*. Conversely, the dry season, from October to March of each year, is often the overwintering pupa stage for *T. aureus* [43]. Therefore, it is speculated that the temperature during this period is closely related to the formation and maintenance of the overwintering pupae of *T. aureus*.

As is well known, *T. aureus* inhabits high-altitude mountainous regions in southeastern China [31,35]. Therefore, altitude is also an important factor contributing to the current potential suitable distribution for *T. aureus*. Based on the predicted potential suitable distribution, the altitude niche ranges from 500 to 1500 m (Appendix A), which is consistent with the current field observation records [31,35]. However, compared to the Maxent model, the RF model revealed a significant expansion in the areas of the medium and low suitability zones, and it showed a large area of continuous distribution (Figure 4b), which seemed illogical. Therefore, we believed that the distribution of medium and low suitability zones under the RF model was controversial. Fortunately, the distribution of high suitability zones was consistent between the Maxent and RF models, and it aligned with the current field survey records [28,29,30].

It is widely believed that with climate change, species tend to move towards higher latitudes or altitudes to seek suitable climatic niches [83,84,85]. However, our results showed that the future altitude niche of *T. aureus* experienced no significant change (Appendix A), but there was a slight increase in the altitude niche of its host plants (Appendix A). Therefore, it could be seen that *T. aureus* also exhibited extremely stable altitude specificity. With climate change, the climatic conditions within its altitude niche became increasingly unfavorable, leading to a lower probability of distribution, implying that its future survival status was becoming more and more precarious. However, another explanation is that there was a hysteresis effect in the response of *T. aureus* to climate change. The host plants are the primary factor influencing the distribution of *T. aureus* (Appendix A). With climate change, the altitude niche of the host plants gradually increased (Appendix A), and it needed to reach a certain level before it would prompt *T. aureus* to follow the host plants and exhibit changes in its altitude niche. The hysteresis effect of the higher trophic level species to climate change is caused by biotic interactions and requires longer time scales to become evident.

Lastly, elevation and climate factors not only directly impacted *T. aureus* but could also exert indirect effects through its host plants (Figure 3). The duality of abiotic factors affecting a species or taxa in a higher trophic level due to biotic interactions is widespread among herbivorous insects [32,75]. As primary producers, the distribution of plants is primarily influenced by abiotic factors [86]. For herbivorous insects, especially those with a high host specialization, their distribution is also closely related to abiotic factors, but they are primarily determined by their preferred host plants. The impact of the abiotic factors on a species across different trophic levels may exhibit the hysteresis effect, as seen in the different responses of *T. aureus* and its host plants to climate change based on their altitude niches. However, the hysteresis effect requires further validation.

### 4.3. Endangered Status and Conservation Recommendations

Based on the AOO and EOO values calculated from the occurrence points and predicted potential suitable distribution data, it was suggested that the current status of *T. aureus* is not threatened. This appeared to coincide with recent reports of the discovery of *T. aureus* in many mountainous regions over the past few years [28,29,30]. Of course, assessing the threatened status of a species solely based on distribution data is controversial and requires consideration of other information, such as the population size and stability, to yield reliable and persuasive results [45]. While the current potential suitable distribution appears large, it is highly fragmented (Figure 4), likely comprising numerous small populations that are vulnerable to climate change and human disturbance, resulting in extremely low stability. For example, the population of *T. aureus hainani* Lee, mainly distributed in Hannan Island [25,87], will gradually disappear with climate change in the future (Appendix A). Overall, the potential suitable distribution of *T. aureus* showed a trend of reduction with climate change, leading to an increasingly threatened level in the future, and gradually moving towards near threatened, vulnerable, or endangered status (Figure 6). Moreover, the vast majority of suitable habitats are located outside protected areas (Appendix A). Therefore, despite the relatively wide distributions, features such as habitat fragmentation, small populations, narrow altitude range, vulnerability to climate change, host plant specialization, and the survival status of its highly dependent host plants, all suggest that its survival status may be more precarious than imagined, so we recommend designating *T. aureus* as currently vulnerable.

How can *T. aureus* be scientifically and effectively protected? Host plants are the most critical factor influencing the distribution of *T. aureus*, and are regional endemic species. Many of them also have features that habitat fragmentation, small populations, and vulnerability to climate change. Therefore, firstly, we need to protect its host plants by prohibiting indiscriminate logging and deforestation. By artificial planting, we can increase the species diversity and abundance of host plants in the habitats of *T. aureus*, providing enough resources. Secondly, further understanding the life habits of *T. aureus* and developing artificial breeding and rearing techniques. Thirdly, raising awareness among the public to prohibit indiscriminate capture and illegal trade

## 5. Conclusions

*Teinopalpus aureus* Mell is a species endemic to high altitudes in southern East Asia, being exceptionally beautiful and rare. Despite its significant conservation value, its spatial distribution remains unclear. Considering its high host specificity, host plants may influence its species distribution. The similar climatic niches further confirm the close relationship between the species and its host plants. This study integrates climate, host plants, and an empirical expert map to predict the potential distribution of *T. aureus* based on the Maxent model. The results indicate that using the species richness of host plants as a surrogate for biotic interactions is a simple and effective way to significantly improve the predictive performance of the SDMs. The current suitable distribution of *T. aureus* and its host plants is highly fragmented, mainly distributed near the Nanling and Wuyi Mountains, and consisting of numerous isolated small populations. With climate change, their distribution areas will significantly shrink, increasing the threatened level in the future. Particularly for the population of *T. aureus* hainani Lee, the likelihood of extinction is extremely high, warranting attention from conservation biologists. Abiotic factors can not only directly affect the distribution of *T. aureus* but also indirectly impact it through its influence on host plants. Compared to its host plants, a phenomenon known as the “hysteresis effect,” caused by biotic interactions, makes the response to climate change of the higher trophic level species take a longer time to become evident. Overall, we tentatively suggest classifying *T. aureus* as a vulnerable species. In the future, multiple measures could be taken to indirectly protect its feeding and habitat resources by conserving host plants, thereby improving the survival status of *T. aureus*.

## Figures and Tables

**Figure 1 insects-15-00879-f001:**
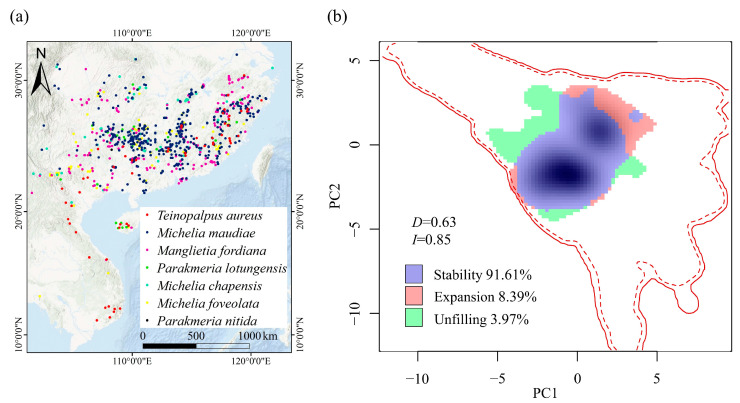
The occurrence points of *T. aureus* and its six host plants (**a**) and their niche overlap (**b**).

**Figure 2 insects-15-00879-f002:**
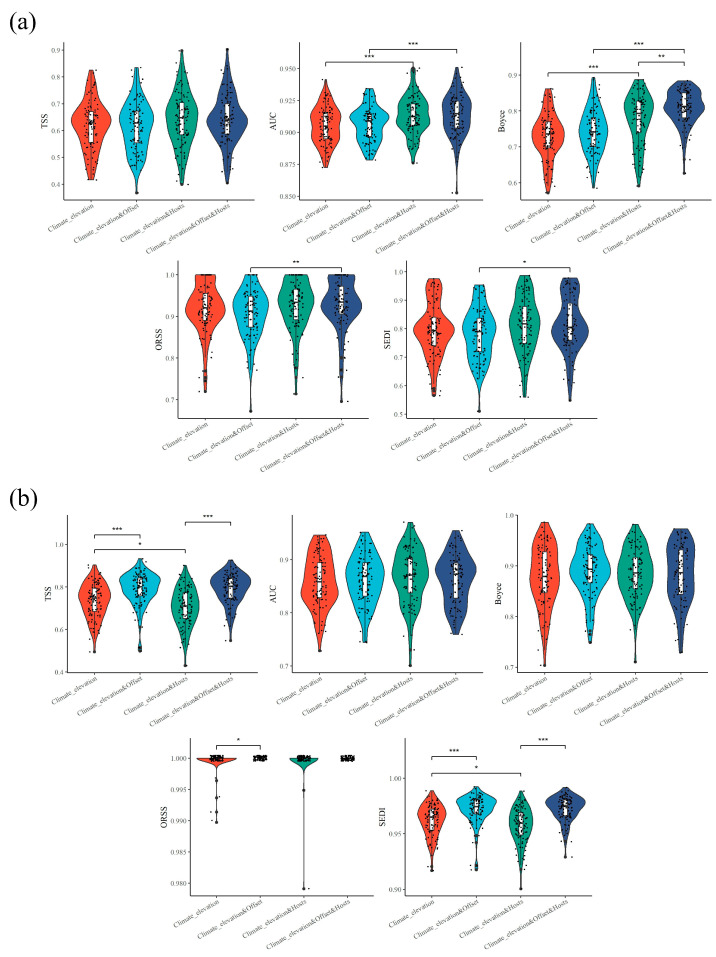
The model performance under four different combinations of predictive variables based on Maxent (**a**) and RF (**b**) models. Significance level: * represents *p* ≤ 0.05, ** represents *p* ≤ 0.01, *** represents *p* ≤ 0.001.

**Figure 3 insects-15-00879-f003:**
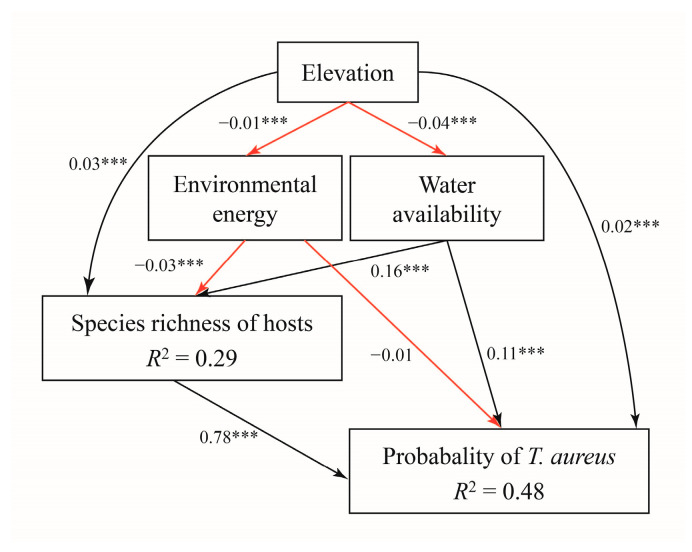
Results of structural equation modeling showing the relative effects of environmental energy, water availability, elevation, and specific host plants on distribution probability of *T. aureus.* Numbers along the arrows (gray, positive; red, negative) represent standardized path coefficients. Significance level: *** represents *p* ≤ 0.001.

**Figure 4 insects-15-00879-f004:**
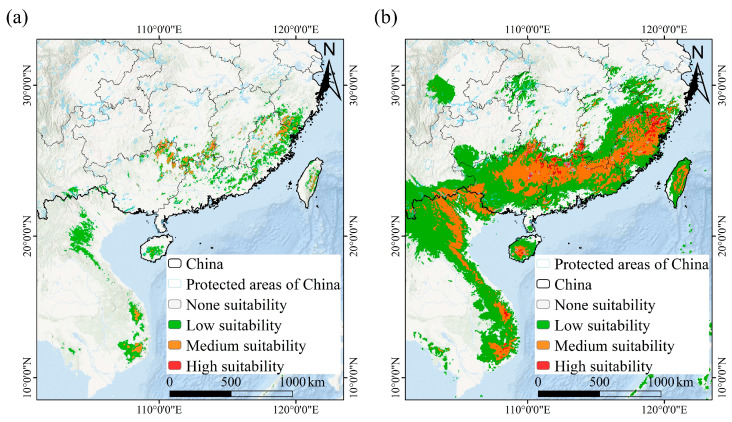
The current potential distribution of *T. aureus* based on the predictive model with combined climatic, elevation, expert map, and species richness of host plants data under Maxent (**a**) and RF (**b**) models.

**Figure 5 insects-15-00879-f005:**
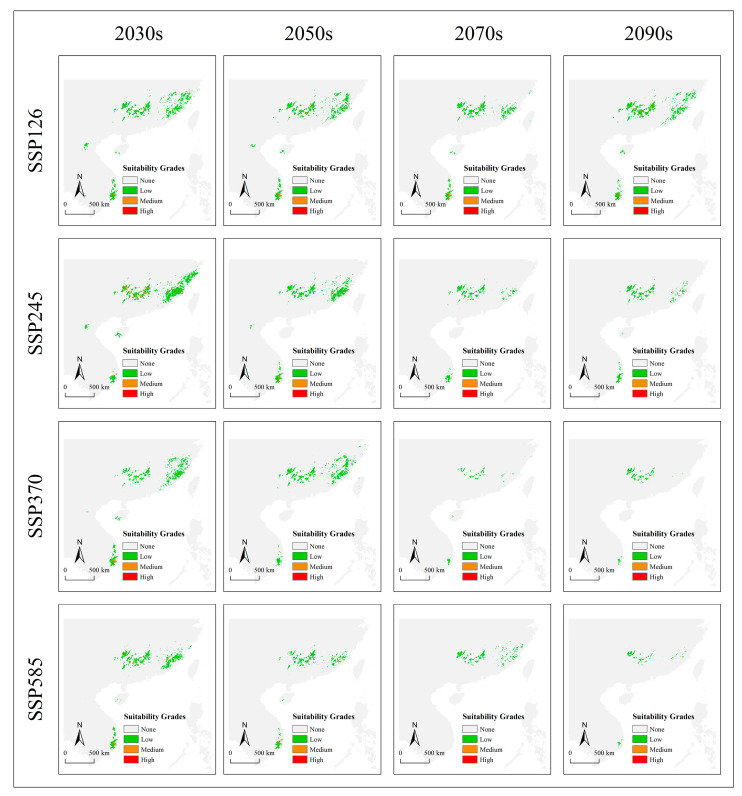
The future potential distribution of *T. aureus* based on the predictive model with combined climatic, elevation, expert map, and species richness of host plants data under Maxentmodel under different climate scenarios.

**Figure 6 insects-15-00879-f006:**
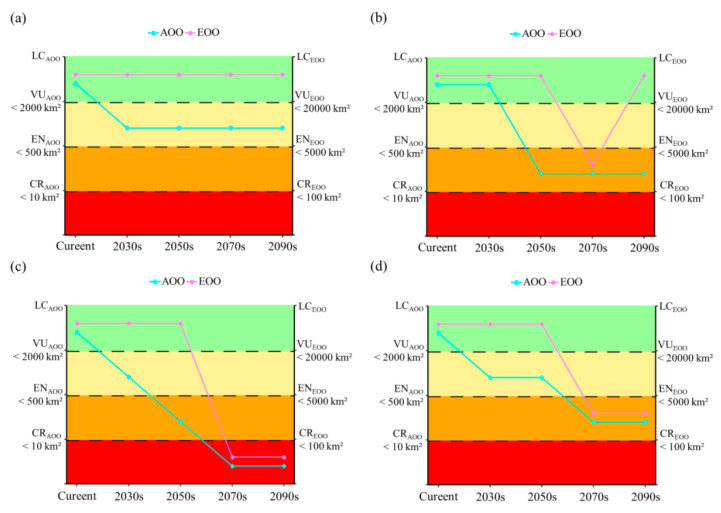
The trend of threatened level for *T. aureus* in the future based on AOO and EOO under different climate scenarios ((**a**): SSP126, (**b**): SSP245, (**c**): SSP370, (**d**): SSP585).

## Data Availability

The data generated during this study are reported in the manuscript.

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
