# Peer review of "Predicting Potential Distribution of Teinopalpus aureus Integrated Multiple Factors and Its Threatened Status Assessment"

_insects, 2024, doi:10.3390/insects15110879_

Round 1
Reviewer 1 Report
Comments and Suggestions for Authors
Review for “Predicting potential distribution of Teinopalpus aureus integrated mutiple factors and its threatened status assessment” by Du et al. (insects-3235201) submitted to the journal Insects to be considered for publication
The manuscript by Du et al. (insects-3235201) presents a species distribution modelling for the butterfly Teinopalpus aureus. The author team used the distribution records of the species from previous publications and modelled future distribution of these species for different scenarios using Maxent modelling algorithms by integrating climate data, host plants, and empirical expert map to predict potential distribution. Their results indicated that utilizing species richness of host plants as a surrogate for biotic interaction was a simple and effective way to significantly improve the predictive preference of SDMs. Based on the modelling results, they tentatively suggested T. aureus.
The manuscript is well arranged, presentation is clear and adequate and the language is easily understandable. However, modelling distribution of a species by application of Maxent to existing distribution data is a routine work (something like examining blood of a patient in a hospital), thus, produce results with little scientific value. Further, as already indicated in the manuscript itself such analyses are sensitive to several probabilities (such as incorrect locality records, parameter setting during analyses ect.) and may produce mislead results. Species distribution modelling can be used as an accessory analysis to support the empirically obtained data. Otherwise, species distribution modelling alone becomes a “virtual science”. Due to this, there is no clear hypothesis in the study. More importantly, a similar study with content of SDM of the species T. aureus (Xing et al. 2019) already has been published, and this publication significantly reduce the originality of the manuscript. Under these statements, I think the present manuscript does not deserve publishing with present content, especially in a high impact journal. Because of insufficient content, there is no sense to suggest some revisions, and rejecting without further consideration is the most appropriate decision.
Author Response
Dear Reviewer,
Thank you very much for the time and effort that you have put into reviewing this manuscript. We really appreciate all your comments and suggestions which have enabled us to improve our work. We have substantially revised the manuscript according to instructions and comments provided in your letter for a re-submission.
Appended to this letter is our point-by-point response to each comment and suggestion raised by you in a red color.
We hope that the revised manuscript is available for considering publication in the Insects. If you have any more comments, please do not hesitate to let us know, we will be much appreciated.
Sincerely yours,
Congcong Du, Gexia Qiao
On behalf of all coauthors,
Institute of Zoology, Chinese Academy of Sciences
Comment 1. The manuscript by Du et al. (insects-3235201) presents a species distribution modelling for the butterfly Teinopalpus aureus. The author team used the distribution records of the species from previous publications and modelled future distribution of these species for different scenarios using Maxent modelling algorithms by integrating climate data, host plants, and empirical expert map to predict potential distribution. Their results indicated that utilizing species richness of host plants as a surrogate for biotic interaction was a simple and effective way to significantly improve the predictive preference of SDMs. Based on the modelling results, they tentatively suggested T. aureus.
Response 1: Thank you.
Comment 2. The manuscript is well arranged, presentation is clear and adequate and the language is easily understandable.
Response 2: Thank you very much for your positive feedback on our manuscript.
Comment 3. However, modelling distribution of a species by application of Maxent to existing distribution data is a routine work (something like examining blood of a patient in a hospital), thus, produce results with little scientific value.
Response 3: Very good comments! Indeed, modeling the distribution of a species by applying Maxent to existing distribution data is routine work, widely used by many researchers, resulting in a lot of similar work. However, this also proves the simplicity and effectiveness of the Maxent model in predicting species distribution.
Additionally, in our manuscript, we were not simply using the Maxent model for prediction but were more focused on how to incorporate biotic interaction into SDMs to improve their predictive performance. Based on our previously proposed hypothesis (the resource dependent hypothesis, Du et al., 2020), we speculated that for specialized herbivorous insects, incorporating the host plant species richness into SDMs might enhance their predictive performance. To test this hypothesis, we focused on Teinopalpus aureus, one representatively protected insect, using the Maxent model to explore the question: Could biotic interactions enhance the predictive performance of SDMs, which was our primary scientific concern.
Comment 4. Further, as already indicated in the manuscript itself such analyses are sensitive to several probabilities (such as incorrect locality records, parameter setting during analyses ect.) and may produce mislead results.
Response 4: Great minds think alike. Your concern is very similar to that of another reviewer. Therefore, following the suggestion of the other reviewer, we re-analyzed the data using a random forest (RF) model with background data, which could handle complex, non-linear interactions and high-dimensional data, and often provided high predictive accuracy. The results showed that:
- After adding the host plants and expert map, the predictive performance of the SDMs improved to a certain extent(Table 1). Notably, the improvement in the TSS and SEDI evaluation metrics was statistically significant (Figure 1).
- Compared to the maxent model, the high suitability areas for aureusrevealed by the RF model were very similar, both mainly distributed in the Nanling Mountains located at the border of Guangxi, Guangdong, Hunan, and Jiangxi provinces, and in mountainous areas at the border of Fujian, Zhejiang, and Jiangxi provinces, such as the Yandang Mountains, Jiufeng Mountains, and Wuyi Mountains (Figure 2).
However, compared to the MaxEnt model, the RF model revealed a significant expansion in the area of medium and low suitability zones (Figure 2). Based on the currently recorded distribution areas of T. aureus and our understanding of its biological characteristics, this seemed illogical. For instance, T. aureus was distributed at mid-to-high altitudes, which meant that this species would certainly showed a fragmented distribution in central or southern China. Yet, the RF model revealed a large area of continuous distribution. Therefore, we believed that the distribution of medium and low suitability zones in this result was controversial. Fortunately, the distribution of high suitability zones was consistent with the RF model and aligns with current field survey data.
(3) With climate change, similar to the maxent model, the potential suitable habitat range of T. aureus also gradually decreased under different greenhouse gas concentration scenarios (Figure 3), manifested by a reduction in remaining areas and an increase in shrinking regions.
In conclusion, although different models might produce minor discrepancies, they all revealed above similar results, which to some extent confirms the accuracy and reliability of the results in our manuscript produced by Maxent model.
Table 1 The average values of different evaluation metrics for the RF model under various combinations of predictor variables.
|
Model |
AUC |
TSS |
boyce |
SEDI |
ORSS |
|
Climate_elevation |
0.858 |
0.736 |
0.879 |
0.962 |
0.999 |
|
Climate_elevation_Offset |
0.862 |
0.789 |
0.889 |
0.971 |
1.000 |
|
Climate_elevation_Host |
0.868 |
0.707 |
0.885 |
0.957 |
0.999 |
|
Climate_elevation_Host_Offset |
0.863 |
0.792 |
0.882 |
0.973 |
1.000 |
Figure 1 The model performance under four different combinations of predictive variables.
Figure 2 The current potential distribution of T. aureus based on the predictive model combined climatic, elevation, expert map and the species richness of host plants data.
Figure 2 The trend of future potential suitable areas (a) and changes (b) of T. aureus under different climate scenarios
Comment 5. Species distribution modelling can be used as an accessory analysis to support the empirically obtained data. Otherwise, species distribution modelling alone becomes a “virtual science”. Due to this, there is no clear hypothesis in the study.
Response 5: Thanks for your comments. Indeed, SDM is just one method. In our research, we just used this method to verify our hypothesis: for specialized herbivorous insects, incorporating the host plant species richness into SDMs might enhance their predictive performance. Moreover, based on this, we have also proposed the "hysteresis effect" caused by biotic interaction, namely, the change in the altitudinal niche of higher trophic level species in response to climate change requires longer time scales to become evident.
Additionally, in recent years, distribution records of T. aureus have been found in many high suitable areas predicted by our manuscript (Meng et al., 2016; Huang & Mao, 2016; Zheng, 2022), which not only proves the reliability of the prediction but also provides a reference for fully understanding the true distribution of this species in the future.
Comment 6. More importantly, a similar study with content of SDM of the species T. aureus (Xing et al. 2019) already has been published, and this publication significantly reduce the originality of the manuscript.
Response 6: Thanks for your comments. In our INTRODUCTION section, we also reviewed the research of Xing et al. (2019) that she predicted the suitable habitats of T. aureus taking both climate suitability and habitat type into account by refining its distribution map to areas with forest cover information. However, despite its current occurrences seemed to depend on evergreen broad-leaved forests, it showed high host specialization, mainly hosted on some species in Magnoliaceae, especially during the egg and larval stages (Wang et al., 2022). The “resource-dependent hypothesis” also stated that the intricate relationship with host plants for resources might constrain their species diversity pattern from being positively correlated for specialist herbivorous insects (Du et al., 2020). Wang et al. (2022) also found that this butterfly exclusively occurred in the preferred primeval broadleaf forests, mainly driven by the specific larval hostplants (i.e., three Magnoliaceae species). Thus it could be seen that only by integrating real host plant information could we accurately predict the distribution of T. aureus (P3L80-P3L91).
Comment 7. Under these statements, I think the present manuscript does not deserve publishing with present content, especially in a high impact journal. Because of insufficient content, there is no sense to suggest some revisions, and rejecting without further consideration is the most appropriate decision.
Response 7: Thanks for your comments. Although maxent model is a very conventional approach, we found, based on this method (including the RF model), that incorporating host plant species richness into SDMs for specialized herbivorous insects could improve their predictive performance. This indicates that utilizing species richness of host plants as a surrogate for biotic interactions is a simple and effective way to significantly enhance the predictive performance of SDMs.
On this basis, by integrating climate data, host plants, and empirical expert maps to predict potential distribution, we accurately predicted the current and future potential suitable habitats for the T. aureus. This provides a foundation for accurately understanding the distribution of this species and, for the first time, assesses its endangered status, laying a scientific foundation for its protection and management.
More importantly, we halso proposed the "hysteresis effect" caused by biotic interaction, namely, the change in the altitudinal niche of higher trophic level species in response to climate change requires longer time scales to become evident.
Finally, we hope you for giving us a opportunity to improve our manuscript for its potential publication in Insects.
Thanks again for all these comments and suggestion! Hopefully we have addressed all concerns.
All my best wishes,
Congcong Du, Gexia Qiao
On behalf of all coauthors
2024/10/14
Institute of Zoology, Chinese Academy of Sciences, Key Laboratory of Zoological Systematics and Evolution
No.1 Beichen West Road, Chaoyang District, Beijing 100101, China
E-mail: Cleverduwang@gmail.com

Reviewer 2 Report
Comments and Suggestions for Authors
Predicting potential distribution of Teinopalpus aureus integrated mutiple factors and its threatened status assessment
Title: Please check the word “mutiple”, I think it should be multiple
General comments:
1. Several instances scientific names are not italicized
2. The manuscript refers to the species with both its common name (Golden Kaiser-I-Hind) and scientific name (Teinopalpus aureus) inconsistently throughout. Ensure consistent usage of the species name.
3. Hypothesis for the study should be properly defined.
4. Satisfactory with the methodology followed.
5. The manuscript uses commas incorrectly or lacks appropriate punctuation in several places, such as after the word "therefore" in line 67.
6. Good research in ecological perspective.
Line 17: The term "predictive preference" is unclear. Instead "predictive performance," would be better.
Line 32: should be corrected to "is highly fragmented" since the subject is "distribution," which is singular
Line 36: Rephrase as "Abiotic factors not only directly affect the distribution."
Line 41: Rephrase as "enhancing the survival prospects."
Line 48: Simply rephrase as "Predicting the niche and potential distribution of species is essential for biodiversity research."
Line 64: change to “biotic variables are less relevant”
Line 70: The study primarily focused on the Golden Kaiser-I-Hind (Teinopalpus aureus), but did not address its ecological role, aesthetic value, or other significant aspects. For example, if the populations being studied require conservation efforts, the benefits to the ecosystem or humanity should be discussed in the introduction or presented as a hypothesis for the research.
Line 71: I don’t think, mentioning of class and phylum is necessary
Line 89: give space between host plants
Line 90, 285: Italicize T. aureus
Line 140: the procedure is confusing, rewrite
Line 196, 218: Check the reference format
Line 244: Refer the following manuscript for detailed assessment “Raghavendra, K.V., Bhoopathi, T., Gowthami, R., Keerthi, M.C., Suroshe, S.S., Ramesh, K.B., Thammayya, S.K., Shivaramu, S. and Chander, S., 2022. Insects: biodiversity, threat status and conservation approaches. Current Science, 122(12), pp.1374-1384.
Line 361: Rephrase: importance of host plants for T. aureus distribution.
Line 464: Rephrase: The Golden Kaiser-I-Hind (Teinopalpus aureus Mell) is a species endemic to
Line 467: Rephrase: The similar climatic niches further confirm the close relationship between the species and its host plants
Line 469: insert “an” before empirical
Line 480: Rephrase: compared to its host plants, a phenomenon known as the 'hysteresis effect,'
Line 481: Rephrase: We tentatively suggest classifying T. aureus as a vulnerable species
Comments on the Quality of English LanguageIt requires improvement
Author Response
Dear Reviewer,
Thank you very much for the time and effort that you have put into reviewing this manuscript. We really appreciate all your comments and suggestions which have enabled us to improve our work. We have substantially revised the manuscript according to instructions and comments provided in your letter for a re-submission.
Appended to this letter is our point-by-point response to each comment and suggestion raised by you in a red color.
We hope that the revised manuscript is available for considering publication in the Insects. If you have any more comments, please do not hesitate to let us know, we will be much appreciated.
Sincerely yours,
Congcong Du, Gexia Qiao
On behalf of all coauthors,
Institute of Zoology, Chinese Academy of Sciences
Comment 1. Title: Please check the word “mutiple”, I think it should be multiple.
Response 1: Thank you. Very sorry for our mistake, we have revised it in our revised manuscript (P1L3).
Comment 2. Several instances scientific names are not italicized.
Response 2: Thank you. Very sorry for our mistake, we have revised them throughout the manuscript the manuscript.
Comment 3. The manuscript refers to the species with both its common name (Golden Kaiser-I-Hind) and scientific name (Teinopalpus aureus) inconsistently throughout. Ensure consistent usage of the species name.
Response 3: Thanks for your suggestion. In order to avoid confusion and misunderstanding, we have standardized the use of scientific names throughout the manuscript.
Comment 4. Hypothesis for the study should be properly defined..
Response 4: Thanks for your suggestion. In the last sentence of the third paragraph in INTRODUCTION section, we proposed the hypothesis: integrating real host plant information into SDMs can enhance their prediction performance (P3L92-P3L93).
Comment 5. The manuscript uses commas incorrectly or lacks appropriate punctuation in several places, such as after the word "therefore" in line 67.
Response 5: Thanks for your comments. Very sorry for our mistake, we have revised them throughout the manuscript.
Comment 6. Good research in ecological perspective.
Response 6: Thank you very much for your positive feedback on our manuscript.
Comment 7. Line 17: The term "predictive preference" is unclear. Instead "predictive performance," would be better.
Response 7: Thanks for your suggestion. We have revised it throughout the manuscript.
Comment 8. Line 32: should be corrected to "is highly fragmented" since the subject is "distribution," which is singular.
Response 8: Thank you. Very sorry for our mistake, we have revised it (P1L36).
Comment 9. Line 36: Rephrase as "Abiotic factors not only directly affect the distribution."
Response 9: Thank you. We have revised it (P1L39-P1L40).
Comment 10. Line 41: Rephrase as "enhancing the survival prospects."
Response 10: Thank you. We have revised it (P1L44-P1L45).
Comment 11. Line 48: Simply rephrase as "Predicting the niche and potential distribution of species is essential for biodiversity research."
Response 11: Thank you. We have revised it (P2L51-P2L52).
Comment 12. Line 64: change to “biotic variables are less relevant”
Response 12: Thank you. We have revised it (P2L66).
Comment 13. Line 70: The study primarily focused on the Golden Kaiser-I-Hind (Teinopalpus aureus), but did not address its ecological role, aesthetic value, or other significant aspects. For example, if the populations being studied require conservation efforts, the benefits to the ecosystem or humanity should be discussed in the introduction or presented as a hypothesis for the research.
Response 13: Thank you. In the INTRODUCTION section of revised manuscript, we have supplemented descriptions regarding the ecological value and other aspects of the golden-spotted swallowtail butterfly (P3L75-P3L78).
Comment 14. Line 71: I don’t think, mentioning of class and phylum is necessary.
Response 14: Thank you. We have deleted them (P3L72-P3L73).
Comment 15. Line 89: give space between host plants.
Response 15: Thank you. Very sorry for our mistake, we have revised it (P3L91).
Comment 16. Line 90, 285: Italicize T. aureus
Response 16: Thank you. Very sorry for our mistake, we have revised it throughout the manuscript the manuscript.
Comment 17. Line 140: the procedure is confusing, rewrite
Response 17: Thanks for your comments. We have supplemented the procedure that how the expert map was updated (P4L146-P4L149).
Comment 18. Line 196, 218: Check the reference format
Response 18: Very sorry for our mistake. We have revised the corresponding reference format throughout the manuscript the manuscript.
Comment 19. Line 244: Refer the following manuscript for detailed assessment “Raghavendra, K.V., Bhoopathi, T., Gowthami, R., Keerthi, M.C., Suroshe, S.S., Ramesh, K.B., Thammayya, S.K., Shivaramu, S. and Chander, S., 2022. Insects: biodiversity, threat status and conservation approaches. Current Science, 122(12), pp.1374-1384.
Response 19: Thanks for your suggestion. We carefully read this article several times, which reviewed the insect diversity, threat status, and conservation approaches assessed by the IUCN. The article presented some new perspectives, such as the grouping of threatened species into different categories based on feeding habits, which implies the complexity of assessing species' threat status, and calls for more efforts to estimate insect threat levels to formulate biodiversity policies aimed at improving the status of threatened species, but it did not provide specific evaluation criteria. Therefore, in our manuscript, we still use the IUCN standards to assess the threat status of T. aureus.
Comment 20. Line 361: Rephrase: importance of host plants for T. aureus distribution.
Response 20: Thank you. We have revised it (P10L355- P10L356).
Comment 21. Line 464: Rephrase: The Golden Kaiser-I-Hind (Teinopalpus aureus Mell) is a species endemic to
Response 21: Thank you. We have revised it (P12L471).
Comment 22. Line 467: Rephrase: The similar climatic niches further confirm the close relationship between the species and its host plants
Response 22: Thank you. We have revised it (P12L474-P12L475).
Comment 23. Line 469: insert “an” before empirical
Response 23: Thank you. We have revised it (P12L476).
Comment 24. Line 480: Rephrase: compared to its host plants, a phenomenon known as the 'hysteresis effect,'
Response 24: Thank you. We have revised it (P12L486-P12L488).
Comment 25. Line 481: Rephrase: We tentatively suggest classifying T. aureus as a vulnerable species
Response 25: Thank you. We have revised it (P12L488- P12L489).
Thanks again for all these comments and suggestion! Hopefully we have addressed all concerns.
All my best wishes,
Congcong Du, Gexia Qiao
On behalf of all coauthors
2024/10/14
Institute of Zoology, Chinese Academy of Sciences, Key Laboratory of Zoological Systematics and Evolution
No.1 Beichen West Road, Chaoyang District, Beijing 100101, China
E-mail: qiaogx@ioz.ac.cn

Reviewer 3 Report
Comments and Suggestions for Authors
This is a very important case study in which the authors explore species interactions and expert maps to enhance the predictive performance of SDMs. The study is elegante and as an additional value the possibility of evaluating the IUCN RED LIST status of a rare butterfly.
I have the following suggestions:
1) The authors argue that MAXENT was the only possibility for the performed SDMs. However, Maxent is a presence-background method and only provides estimates of relative suitability regardless of how the background sample is specified. I suggest that at least another method (within the scope of Bioclim (Bioclimatic Envelope Modeling) can be used to verify the consistency of the obtained results. One possibility is the use of Random Forest (RF) with Background Data, that can handle complex, non-linear interactions and high-dimensional data. Often provides high predictive accuracy.
2) The authors should provide in the Introduction more publications in which the host plants are used as predictors in SDMs
Detailed corrections
-In lines 90, 115 and 117, T. aureus should be in italic
Author Response
Dear Reviewer,
Thank you very much for the time and effort that you have put into reviewing this manuscript. We really appreciate all your comments and suggestions which have enabled us to improve our work. We have substantially revised the manuscript according to instructions and comments provided in your letter for a re-submission.
Appended to this letter is our point-by-point response to each comment and suggestion raised by you in a red color.
We hope that the revised manuscript is available for considering publication in the Insects. If you have any more comments, please do not hesitate to let us know, we will be much appreciated.
Sincerely yours,
Congcong Du, Gexia Qiao
On behalf of all coauthors,
Institute of Zoology, Chinese Academy of Sciences
Comment 1. This is a very important case study in which the authors explore species interactions and expert maps to enhance the predictive performance of SDMs. The study is elegante and as an additional value the possibility of evaluating the IUCN RED LIST status of a rare butterfly.
Response 1: Thank you very much for your positive feedback on our manuscript.
Comment 2. The authors argue that MAXENT was the only possibility for the performed SDMs. However, Maxent is a presence-background method and only provides estimates of relative suitability regardless of how the background sample is specified. I suggest that at least another method (within the scope of Bioclim (Bioclimatic Envelope Modeling) can be used to verify the consistency of the obtained results. One possibility is the use of Random Forest (RF) with Background Data, that can handle complex, non-linear interactions and high-dimensional data. Often provides high predictive accuracy..
Response 2: Very good comments! Followed your suggestion, we re-analyzed the data using the random forest (RF) mode. The results showed that:
- After adding the host plants and expert map, the predictive performance of the SDMs improved to a certain extent(Table 1). Notably, the improvement in the TSS and SEDI evaluation metrics was statistically significant (Figure 1).
- Compared to the maxent model, the high suitability areas for aureusrevealed by the RF model were very similar, both mainly distributed in the Nanling Mountains located at the border of Guangxi, Guangdong, Hunan, and Jiangxi provinces, and in mountainous areas at the border of Fujian, Zhejiang, and Jiangxi provinces, such as the Yandang Mountains, Jiufeng Mountains, and Wuyi Mountains (Figure 2).
However, compared to the MaxEnt model, the RF model revealed a significant expansion in the area of medium and low suitability zones (Figure 2). Based on the currently recorded distribution areas of T. aureus and our understanding of its biological characteristics, this seemed illogical. For instance, T. aureus was distributed at mid-to-high altitudes, which meant that this species would certainly showed a fragmented distribution in central or southern China. Yet, the RF model revealed a large area of continuous distribution. Therefore, we believed that the distribution of medium and low suitability zones in this result was controversial. Fortunately, the distribution of high suitability zones was consistent with the RF model and aligns with current field survey data.
(3) With climate change, similar to the maxent model, the potential suitable habitat range of T. aureus also gradually decreased under different greenhouse gas concentration scenarios (Figure 3), manifested by a reduction in remaining areas and an increase in shrinking regions.
In conclusion, although different models might produce minor discrepancies, they all revealed above similar results, which to some extent confirms the accuracy and reliability of the results in our manuscript produced by Maxent model.
Table 1 The average values of different evaluation metrics for the RF model under various combinations of predictor variables.
|
Model |
AUC |
TSS |
boyce |
SEDI |
ORSS |
|
Climate_elevation |
0.858 |
0.736 |
0.879 |
0.962 |
0.999 |
|
Climate_elevation_Offset |
0.862 |
0.789 |
0.889 |
0.971 |
1.000 |
|
Climate_elevation_Host |
0.868 |
0.707 |
0.885 |
0.957 |
0.999 |
|
Climate_elevation_Host_Offset |
0.863 |
0.792 |
0.882 |
0.973 |
1.000 |
Figure 1 The model performance under four different combinations of predictive variables.
Figure 2 The current potential distribution of T. aureus based on the predictive model combined climatic, elevation, expert map and the species richness of host plants data.
Figure 2 The trend of future potential suitable areas (a) and changes (b) of T. aureus under different climate scenarios
Comment 3. The authors should provide in the Introduction more publications in which the host plants are used as predictors in SDMs.
Response 3: Thanks for your suggestion. In the INTRODUCTION section, we appropriately supplemented recent reviews, calls, and attempts to incorporate biotic interactions into SDMs (P3L92-P3L93).
Comment 4. In lines 90, 115 and 117, T. aureus should be in italic.
Response 4: Thank you. Very sorry for our mistake, we have revised it in our revised manuscript.
Thanks again for all these comments and suggestion! Hopefully we have addressed all concerns.
All my best wishes,
Congcong Du, Gexia Qiao
On behalf of all coauthors
2024/10/14
Institute of Zoology, Chinese Academy of Sciences, Key Laboratory of Zoological Systematics and Evolution
No.1 Beichen West Road, Chaoyang District, Beijing 100101, China
E-mail: qiaogx@ioz.ac.cn

Round 2
Reviewer 1 Report
Comments and Suggestions for Authors
In first round of the review, decision was made based on main points regarding data content of the manuscript. The first was related to scientific contribution provided by species distribution modelling. Species distribution has little scientific value without empirical data. Species distribution modelling alone is a “virtual science” or a “statistical game”, nothing more. There is nothing new in this respect in the revised version of the manuscript. The second and the more important issue was existing of a published study with largely overlapping content. As cited in the manuscript Xing et al. (2019) conducted a species distribution modelling on the same species T. aureus, and this reduce originality value significantly. The responds by author team do not change these facts, thus there is no new reason to make a new decision. Even I think, for such cases the editor should not ask for second time to reviewer to re-check their decision.